# Synergistic Effects of *Lavandula angustifolia* and a Bacterial Consortium on Bioremediation of a Heavy Metal-Contaminated Soil

**DOI:** 10.3390/plants14172734

**Published:** 2025-09-02

**Authors:** Cristina Cavone, Aurora Rutigliano, Pietro Cotugno, Ludovica Rolando, Chiara De Carolis, Anna Barra Caracciolo, Paola Grenni, Ilaria Savino, Antonio Bucci, Gino Naclerio, Fulvio Celico, Vito Felice Uricchio, Valeria Ancona

**Affiliations:** 1National Research Council, Water Research Institute, 70132 Bari, Italy; cristina.cavone@ba.irsa.cnr.it; 2Department of Biosciences and Territory, University of Molise, 86090 Pesche, Italy; antonio.bucci@unimol.it (A.B.); naclerio@unimol.it (G.N.); 3National Research Council, Institute for Construction Technologies, 70124 Bari, Italy; a.rutigliano42@phd.uniba.it (A.R.); ilariasavino@cnr.it (I.S.); vitofelice.uricchio@cnr.it (V.F.U.); 4Department of Soil, Plant and Food Sciences, University of Bari “Aldo Moro”, Via G. Amendola 165/A, 70126 Bari, Italy; 5Department of Chemistry, University of Bari, 70126 Bari, Italy; pietro.cotugno@uniba.it; 6National Research Council, Water Research Institute, Montelibretti, 00010 Rome, Italy; ludovica.rolando@irsa.cnr.it (L.R.); chiara.decarolis@cnr.it (C.D.C.); anna.barracaracciolo@irsa.cnr.it (A.B.C.); paola.grenni@irsa.cnr.it (P.G.); 7Department of Chemistry, Life Sciences, and Environmental Sustainability, University of Parma, 43124 Parma, Italy; fulvio.celico@unipr.it

**Keywords:** plant-assisted bioremediation, lavender, microbial consortium, soil quality index

## Abstract

Heavy metal (HM) contamination represents a significant threat to soil functionality and ecosystem health. The present study aims to assess the efficacy of an integrated bioremediation strategy combining *Lavandula angustifolia* (lavender) and a four-strain bacterial consortium in restoring a multi-contaminated soil collected from a former industrial site in Southern Italy. Microcosm experiments were conducted over a period of 90 days, including three treatments and a control: a planted condition (PLANT), a bioaugmented condition (BIOAUG), and a combined plant and bioaugmentation condition (PLANT+BIOAUG). The control (HCS) consisted of unplanted and non-bioaugmented soil. Soil physico-chemical parameters (e.g., pH, electrical conductivity, and heavy metal concentrations), plant growth, microbial abundance, and dehydrogenase activity (DHA) were measured at the initial and final experimental time. Finally, a Soil Quality Index (SQI) was applied. The combined treatment (PLANT+BIOAUG) promoted a significant reduction in total Pb and Sn concentrations by 44.7% and 66.9%, respectively. Moreover, a significant increase in soil pH and microbial abundance was observed. Applying the SQI to integrate overall soil data made it possible to highlight the highest quality score (0.73) for this condition. These findings suggest the potential effectiveness of lavender-assisted bioaugmentation as a scalable and multifunctional strategy for remediating heavy metal-contaminated soils, in line with ecological restoration principles.

## 1. Introduction

Heavy metal (HMs) pollution is one of the most widespread and hazardous threats to the environment, food security, and ecosystem functioning [1,2], representing a substantial threat to plant growth and productivity [3]. High levels of these elements primarily originated from anthropogenic activities such as metallurgy, mining, industrial production, vehicular traffic, and intensive use of fertilisers and pesticides, and they tend to accumulate in soils. Moreover, they can enter food chains through plant uptake, raising concerns for both environmental and public health [4,5,6]. Unlike organic pollutants, HMs are not biodegradable and can persist in environmental compartments over long periods [7,8], altering chemical and physical properties of soils and compromising biogeochemical cycles [8]. Heavy metals are typically classified as either essential or non-essential nutrients. For example, copper (Cu), zinc (Zn), iron (Fe), manganese (Mn), and molybdenum (Mo) are essential micronutrients essential micronutrients required in minimal quantities by plants to facilitate proper growth and maintain overall health and well-being. Indeed, they fulfil a pivotal role as cofactors for a variety of enzymes, thereby playing a crucial part in fundamental physiological processes [9]. The most critical elements include lead (Pb), zinc (Zn), tin (Sn), cadmium (Cd), nickel (Ni), and chromium (Cr), which are known for their toxicity, mutagenicity, and ability to interfere with essential cellular processes such as photosynthesis, respiration, and DNA replication [5,10]. Traditional soil remediation technologies—such as excavation, chemical washing, and solidification/stabilisation—can be effective, but at the same time, they are costly, invasive, and unsustainable, particularly for large or multi-source contaminated sites [4,11]. These techniques frequently decrease soil quality, reducing microbial biodiversity and hindering long-term ecological recovery [8,12]. Consequently, scientific interest has increasingly shifted towards bioremediation technologies, which utilise plants and microorganisms for removing, detoxifying, or stabilising contaminants. Among these, plant-assisted remediation (including rhizoremediation) has emerged as a promising approach with a low environmental impact, combining purification efficiency with soil regeneration [5,7,13]. This technology employs selected plant species which can absorb, accumulate, stabilise or transform contaminants through differentiated physiological and biochemical mechanisms [14,15], which also involve their associated microbiota [16]. Aromatic and medicinal plants (MAPs), such as *Lavandula angustifolia*, have been identified as being particularly effective in remediation of contaminated soils [17,18]. *L. angustifolia* is an aromatic, perennial shrub native of the Mediterranean area. It is widely cultivated for extraction of its essential oil [19] and it is known for combining several advantageous traits [20]: tolerance to abiotic stress, low translocation of heavy metals to the epigeal part and to oil fractions, and stimulation of beneficial rhizo-microbiota through root exudates [21] when cultivated on polluted soils [20,22]. Recent studies have shown that when lavender is combined with soil amendments such as biochar [23] or with the bioaugmentation [24], its ability to accumulate and stabilise lead (Pb), cadmium (Cd), nickel (Ni), and zinc (Zn) is enhanced.

Overall characteristics make lavender a promising candidate for integrated phytoremediation approaches. Beyond its remediation capacity, the utilisation of lavender in urban landscapes can be further supported by its aesthetic appeal and drought tolerance, making it well suited for incorporation into green infrastructure in contaminated or degraded peri-urban areas. Furthermore, its cultivation makes it possible biomass valorisation through extraction of essential oils or production of compost, biochar, and hydrolats, thereby reducing risks of contaminants entering food chains [25,26].

Nevertheless, phytoremediation can be less effective in heavily contaminated soils, where metal bioavailability is low and microbial activity is low. To address these limitations, microbial bioaugmentation—introducing metal-tolerant bacterial consortia—has emerged as a complementary strategy [27]. Microorganisms can transform and immobilise metals through biosorption, bioaccumulation, biotransformation, and redox-mediated precipitation [6,10,28]. Bacterial resistance to heavy metals typically involves a suite of adaptive mechanisms including active efflux through P-type ATPases and RND-type transporters, intracellular sequestration via metallothionein and glutathione, and enzymatic redox transformation of toxic ions [29]. Such mechanisms not only ensure microbial survival in polluted environments but also increase bioavailability and mobility of metals, facilitating their uptake or immobilisation in the rhizosphere [30].

In this context, a consortium comprising *Gordonia amicalis* (Actinomycetota), *Rhodococcus erythropolis* (Actinomycetota), *Acinetobacter puyangensis* and *A. tibetensis* (Pseudomonadota, γ-Protebacteria) strains was tested for improving lavender remediation capacity in removing HMs. These genera are recognised for their biodegradation capacity, including reductive dechlorination and co-metabolic transformation [8]. Additionally, they produce siderophores and biosurfactants which enhance metal chelation and solubilisation of hydrophobic organic compounds, respectively [31,32]. Bacteria belonging to these genera have demonstrated strong biofunctional properties in contaminated environments, particularly in synergy with plant roots [33,34,35]. Positive interactions between plants and rhizo-microbiota have shown to stimulate microbial activity [12], improve soil physico-chemical properties (e.g., pH and electrical conductivity), and promote adaptation to metal stress [12,30,36].

Bacterial consortia are generally more robust than single strains in complex matrices, as they facilitate functional cooperation, extend the range of degradable substrates, and display higher resilience under variable environmental conditions [37,38]. Moreover, biosurfactant-producing bacteria can improve organic and inorganic pollutant bioavailability [39] by lowering surface tension, which is essential for effective breakdown of recalcitrant compounds.

The present study aims to evaluate the combined efficacy of phytoremediation with *Lavandula angustifolia* and bioaugmentation with the microbial consortium (*Gordonia amicalis*, *Rhodococcus erythropolis*, *Acinetobacter puyangensis*, and *Acinetobacter tibetensis* strains) applied to a soil multi-contaminated with Zn, Sn, and Pb from the highly impacted site of Taranto—Cimino Manganecchia. Chemical (pH, electrical conductivity, metal concentrations) and microbiological (total microbial abundance and dehydrogenase activity) investigations were performed and tested with multivariate statistical analyses (ANOVA and PCA) to evaluate treatment effectiveness. Moreover, to assess overall soil improvements, a Soil Quality Index (SQI) was applied, integrating these parameters. The SQI considered heavy metals (Zn, Sn, and Pb) as negative indicators and pH, electrical conductivity (EC), microbial abundance, and dehydrogenase activity as positive indicators, supporting a comprehensive evaluation of soil functionality and remediation effectiveness over time [40]. This approach ensured that both the reduction in pollutants and the restoration of ecological balance were considered in assessing treatment performance.

The overarching objective was to produce empirical data which may support the development of eco-sustainable, scalable, and replicable strategies for remediation of contaminated soils, aligned with bioeconomy principles and long-term restoration of terrestrial ecosystems.

## 2. Results

### 2.1. Plant Growth, Soil pH, and EC

All lavender that was cultivated in the soil microcosms demonstrated robust vitality and exhibited normal development over a period of 90 days. Comparing the PLANT and PLANT+BIOAUG conditions, no significant differences were found in terms of biomass and leaf number. A significant increase in soil pH (*p* < 0.05) compared to the control soil (HCS) was observed in all treated conditions. At 90 days, the PLANT+BIOAUG had the highest value (pH: 7.38 ± 0.07), followed by BIOAUG (pH: 7.28 ± 0.04) and PLANT (pH: 7.13 ± 0.00) ones. The study revealed no statistically significant differences in EC among the treatments examined. The values ranged from 1.03 ± 0.01 to 1.50 ± 0.08 dS/cm. All data are reported in Table 1.

### 2.2. Contaminant Removal and Plant Uptake

Inductively coupled plasma mass spectrometry (ICP-MS) analysis revealed the presence of various heavy metals (Be, Al, Ti, V, Cr, Mn, Fe, Co, Ni, Cu, Zn, As, Se, Sn, Sb, Cd, Pb) in soil samples from the various microcosms. The concentrations of Sn, Zn, and Pb exceeded the Italian legal limits (Italian Legislative Decree 152/06), which are 1 mg/kg for Sn, 150 mg/kg for Zn, and 100 mg/kg for Pb, respectively (Figure 1). However, at 90 days, significant (*p* < 0.05) decreases in Sn concentrations were observed. Specifically, Sn concentrations decreased from 81.01 mg/kg to 26.83 mg/kg in the PLANT+BIOAUG, to 68.89 mg/kg in the PLANT, and to 78.46 mg/kg in the BIOAUG conditions. Moreover, in HCS microcosms, the average Sn concentration observed at 90 days (48.09 mg/kg) was lower than the initial one (81.01 mg/kg). Due to the high variability in Sn concentrations, statistical analysis revealed no significant differences from 0 and 90 days.

No significant decrease in Zn concentration was recorded in all the experimental conditions. Lead (Pb) showed a significant overall decrease (Kruskal–Wallis *p* = 0.02), and concentrations decreased from 1328.98 mg/kg to 841.80 mg/kg in the PLANT, to 1071.99 mg/kg in the BIOAUG, and to 734.31 mg/kg in the PLANT+BIOAUG conditions. As observed for Sn in HCS microcosms, the high variability of Pb values at 90 days did not reveal any significant reduction in Pb concentrations in this experimental condition.

The concentrations of Zn, Sn, and Pb observed in plant tissues (leaves and roots) and the translocation factor (calculated as the ratio of metal concentration in leaves to that in roots) are reported in Table 2. Zn was the most abundant heavy metal found (39.88 ± 16.67 mg/kg).

### 2.3. Microbial Abundance and Activity

The presence of lavender plants (PLANT and PLANT+BIOAUG) in the microcosms resulted in a significant (*p* < 0.001) increase in microbial abundance at 90 days. Differently, in BIOAUG, a sharp decrease from day 0 to 90 d (*p* < 0.05) was observed (Figure 2A). Indeed, the PLANT+BIOAUG condition showed the highest abundance values (up to 1.40 × 10^8^ cells/g), suggesting that the co-presence of the lavender with the bacterial consortium was the best condition for increasing the overall microbial community number and making bacterial consortium survival possible. Conversely, the decline observed in BIOAUG alone may be indicative of limited persistence of the inoculated strains and potential competition with autochthonous microorganisms, which could have hindered consortium establishment.

The PLANT treatment showed the highest dehydrogenase value (DHA: 151.92 μg TPF/g), underscoring the role of plant roots in supporting natural soil microbial communities. DHA values were lower in the PLANT+BIOAUG and in the BIOAUG treatments (Figure 2B).

### 2.4. Soil Quality Index

Figure 3 reports the soil quality index (SQI), which integrated pH, electrical conductivity (EC), microbial abundance (DAPI), and dehydrogenase activity (DHA) values as positive indicators and heavy metal (Zn, Sn, and Pbtot) concentrations as negative ones. The SQI after 90 days shows the highest value in PLANT+BIOAUG (SQI = 0.73), followed by PLANT (SQI = 0.58) and BIOAUG (SQI = 0.48). Interestingly, the control soil (HCS) remained at a very low value (SQI = 0.32). The possibility to use all parameters measured in the SQI made it possible to rank the soil improvement and to give a value of its quality in response to the remediation strategies applied. All parameters, except EC, because its variations were not significant, gave a contribute to the index value. Overall parameters integrated in the SQI demonstrated that the PLANT+BIOAUG condition attained a substantial enhancement in soil quality. The PLANT condition demonstrated a moderate improvement. Differently, bioaugmentation alone was not effective for improving soil quality, presumably because the application of bacterial species which outcompete the natural ones can be negative from an ecological point of view.

### 2.5. Multivariate Analysis

A principal component analysis (PCA) was conducted on the chemical and biological parameters (Figure 4). The first two components, PC1 and PC2, explained 34.1% and 19.9% of the total variance, respectively, for a cumulative contribution of 54.0%. The PC1 shows how heavy metal contamination (Sn and Pb) is in the opposite position of pH and microbial abundance (DAPI), indicating that reductions in metal concentrations were closely associated with improvements in soil microbial growth and neutralisation of acidity. The PC2 shows how DHA and EC are positively correlated, reflecting the influence of root–microbe interactions and enzymatic activity on soil ionic balance. The biplot (Figure 4) demonstrates a clear separation of the treatments over time. Indeed, HCS (control) data exhibited a negative association with PC1, suggesting the presence of residual contamination and lower microbial activity. Considering the treatments after 90 days, PLANT and PLANT+BIOAUG are mainly influenced by overall positive factors (DAPI, DHA, and pH), indicating the collaborative impact of lavender rhizosphere activity and microbial interactions in enhancing soil quality recovery. Furthermore, BIOAUG is not grouped in a cohesive cluster, suggesting the low effectiveness of microbial bioaugmentation alone.

The statistical robustness of the treatment effects on the multivariate soil profiles was confirmed by PERMANOVA, which indicates a significant difference between PLANT+BIOAUG (*p* < 0.001) and PLANT (*p* < 0.05) from the other treatments.

## 3. Discussion

The combined strategy of bioremediation with *Lavandula angustifolia* and bioaugmentation with a selected microbial consortium (PLANT+BIOAUG) has been demonstrated to be effective in decreasing HM contamination (Sn and Pb) and increasing overall soil quality, considering both physical–chemical and microbiological parameters. The increase in soil pH observed across all treatments (particularly in PLANT+BIOAUG) can be attributed to synergistic biological and chemical processes. As demonstrated in the study by Sharma et al. [41], root exudates from *L. angustifolia* interacting with metal ions (e.g., Pb and Sn) reduce proton displacement (H^+^) and increase pH levels. Concurrently, a microbial ammonification process is initiated by the bacterial consortium *(Gordonia, Rhodococcus, Acinetobacter*), resulting in the release of NH_3_ and the concomitant consumption of H^+^ ions [42]. The production of biosurfactants has been demonstrated to enhance metal solubility, thereby promoting precipitation as hydroxides or carbonates at higher pH [43]. In addition, the degradation of acidic metabolites via microbial activity has been shown to contribute to alkalinisation [44]. The interconnected mechanisms underpinning the optimal pH in PLANT+BIOAUG are such that plant–microbe synergies are maximised, thereby facilitating metal immobilisation.

The PLANT treatment showed the highest microbial dehydrogenase value, confirming that lavender presence promoted a general increase in number and activity of natural soil microbial communities, thanks to the positive interactions established in the rhizosphere [12,16,45]. However, DHA values were lower in the bioaugmented conditions (PLANT+BIOAUG and BIOAUG treatments), suggesting competition phenomena between autochthonous and allochthonous bacterial populations. Interestingly, in BIOAUG, the lowest microbial abundance, associated with relatively high microbial activity, can be ascribed to the competitive exclusion of the bacterial consortium. Indeed, the dead bacterial biomass of the inoculants could be a carbon and nutrient source for the autochthonous soil bacteria, stimulating their overall activity [46]. A recent study [47] has demonstrated that the processes of microbial necromass and cell lysis result in the release of dissolved organic carbon and labile nutrients. These nutrients are then rapidly metabolised by native microbial communities. This phenomenon can also explain the low effectiveness of bioaugmentation (BIOAUG) in this experiment, as found in another works [48].

Supporting our results, a recent study on the same consortium used in this experiment has shown that these bacteria are potentially able to tolerate and remove heavy metals and to produce biosurfactants [31]. Other authors have also found that *Acinetobacter* [43] and *Gordonia* [49] genera interact with heavy metals in soil, facilitating their immobilisation and sequestration through various mechanisms such as biosorption, precipitation, and intracellular accumulation, also modulating local metal concentrations in the rhizosphere [29,50]. *Rhodococcus* and *Acinetobacter* have also been reported to produce exosaccharides (EPSs) and biofilm matrices which contribute to HM adsorption and immobilisation through bonds with functional groups such as carboxyls, phosphates, and hydroxyl groups [51]. Moreover, the rhizosphere of *L. angustifolia* has been demonstrated to secrete chelating compounds, including organic acids and phenolic metabolites. These compounds have the capacity to bind HMs and reducing their mobility in soil [52]. Overall, the metabolic characteristics of the genera in the consortium mentioned above, with the natural chelating capacity of lavender root exudates, can explain the best performance of the PLANT+BIOAUG condition, thanks to their active role in HM removal. However, this fact does not exclude a possible contribution of the rhizosphere natural microbial community in HM reduction in the PLANT condition [30].

Our results suggest a pivotal role of *L. angustifolia* in shaping the composition and functionality of the rhizospheric microbial community, which made possible the survival of the microbial consortium and its coexistence with the natural soil microbial populations. This was possible through a combination of chemical and physical modifications, through signalling (e.g., root exudates) and symbiotic interactions, which have defined new rhizosphere microenvironments [53]. Indeed, root exudates comprise a wide variety of chemicals (e.g., simple sugars, organic acids, and amino acids) which enhance microbial activity [41], modify soil chemistry, and contribute to nutrient cycling and acid–base equilibrium [54]. Consequently, lavender favoured Sn and Pb removal, both absorbing/stabilising them, in line with recent work [55] and favouring the microbial role. Any significant reduction in Zn was observed in all the experimental pots. At these Zn values (about 500 mg/kg), lavender with or without the microbial consortium was enabled to promote the decrease in this metal, which is an essential micronutrient for all plant species. In a previous work [20], phytoremediation capability of lavender has been observed at higher concentrations (>2000 mg/kg) than those obtained in the present experiment. In the treatment without lavender and/or bioaugmentation (HCS), the HM results did not show any statistically significant reduction. These results are in line with those obtained in a recent pot experimental trial performed using soil collected from the same survey site of the present work and different plant species (*Brassica juncea* and *Sorghum bicolor*) [56].

The analysis of HMs in leaves and roots evidenced that HM absorption occured primarily via the roots. Zn concentrations observed in plant tissues of the planted microcosms were not in line with the results shown by Ancona et al. [57,58] in previous field phytoremediation experiments. In fact, in this study, any significant Zn uptake was observed in lavender leaves, as evidenced by TF values < 1.

## 4. Materials and Methods

### 4.1. Study Area and Sample Collection

In November 2023, soil samples were collected from a historically contaminated area, located close to Taranto (Apulia Region, Southern Italy; GPS coordinates: 40°28′04.92″ N, 17°18′12.68″ E). The soil was degraded not only in terms of pollution but also in terms of low nutrient content (organic carbon content: 15.02 ± 3.21 g/kg, nitrogen: 0.17 ± 0.07 g/kg and phosphorous: 8.12 ± 1.81 mg/kg), with a pH of 6.64 and EC of 0.99 dS/cm.

This area was classified as a Contaminated Sites of National Interest (SIN) by the Italian government due to long-standing emissions from metallurgical and petrochemical industries. The area has been the subject of plant-assisted bioremediation (PABR) strategies for approximately a decade, with the application of the Monviso poplar clone (*Populus generosa* × *Populus nigra*). A preliminary investigation, conducted by Ancona et al., 2017 [57], on the chemical contamination revealed a non-uniform distribution of polychlorinated biphenyls (PCBs) across various locations within the studied area. The concentrations of PCBs often exceeded the national legal limit (60 ng/g for gardens, parks and residential areas, as outlined by the Italian Legislative Decree 152/06). Additionally, high concentrations of several heavy metals (especially Sn, Pb and Zn) were detected, requiring specific attention.

Approximately 10 kg of a surface a composite soil (0–20 cm) was collected for performing the microcosm experiments. Prior to utilisation, stones, larger clods and residues of plant materials were removed; the soil was then dried at room temperature (18–20 °C) and sieved (2 mm mesh). The soil was put in a cylindrical container and placed on a roller for 3 days for homogenising it.

### 4.2. Plant, Bacterial Consortium, and Experimental Design

*Lavandula angustifolia* (*Lamiaceae* family) is a Mediterranean perennial shrub widely used in phytoremediation for its tolerance to metal-contaminated soils and its ability to support rhizospheric microbial activity [20,24]. *L. angustifolia* plants were obtained from a commercial nursery (Mondo Piante, Terlizzi, Puglia). Healthy and uniform plants (6–7 cm in height) were transplanted into microcosms at the start of the experiment.

The bacterial consortium (previously tested for removal of organic pollutants such as diesel) consisted of the following strains: *Gordonia amicalis* (strain S2S5), *Rhodococcus erythropolis* (strain S2W2), *Acinetobacter puyangensis* (strain S1W1), and *Acinetobacter tibetensis* (strain S2S8). It was obtained as described in Cavone et al. [31], with some modifications. The bacterial cultures were grown separately in 50 mL of LB medium at 28 °C for 24 h until they reached the mid-logarithmic phase. Bacterial cells were then harvested by centrifugation at 3500 rpm for 10 min, washed twice with sterile water, and resuspended in water (10 mL). Each bacterial suspension was adjusted to an optical density (OD 600 nm) of 1.0. Subsequently, equal volumes of the suspensions were mixed to constitute the microbial consortium (10 mL), which has been used in the mesocosm experiments.

The microcosm experiment consisted in plant pots (1 L capacity) containing HM-contaminated soil samples (500 g each). Three different conditions were tested: plant-assisted bioremediation (PLANT), with only a lavender plant, bioaugmentation (BIOAUG) with only the bacterial consortium, and a combination of plant-assisted bioremediation and bioaugmentation (PLANT+BIOAUG). The inoculum was incorporated into the BIOAUG and PLANT+BIOAUG treatments at a final concentration of 2% v/w, with the consortium population being introduced at a concentration of 1 × 10^8^ CFU/g soil. Finally, untreated control microcosms (HCS) consisted exclusively in the historically contaminated soil. Each microcosm was tested in triplicate and maintained in a growth chamber (Climate chamber Binder KBF 720, GmbH, Wetzlar, Germany) at environmental temperature (25 ± 1 °C); photoperiod 16/8 h light/dark; and relative humidity reaching 65/80% light/dark. Microbiological and chemical analyses were performed at the start and at the end of the experiment (90 days). The overall experimental design is illustrated in Figure 5.

### 4.3. Plant Growth Analysis

Plant growth was evaluated at the end the experiment (90 days). The plant height (cm) and the number of leaves (N° leaves) were measured immediately. For the biomass, lavender plants (shoot and roots) were extracted from the soil and weighed. The dry weight was obtained by placing the roots and leaves in an oven at 40 °C until a constant mass was reached.

### 4.4. Physico-Chemical Analysis of the Soil and Pollutant Plant Uptake Assessment

The soil chemical properties, including pH and soil electrical conductivity (EC), were determined in accordance with the Italian Official Methods of Soil Chemistry (MUACS), approved by the Italian Ministry for Agricultural Policies [59]. The pH was determined potentiometrically, following the Method III.1 protocol, on a suspension of soil-neutral salt solution (potassium chloride, KCl) containing 10 g of soil. The electrical conductivity (EC) was carried out following the Method IV.1 protocol, using aqueous soil extracts (5:1 water/soil ratio). The soil extract was subjected to mechanical shaking for 2 h, followed by an equal time of undisturbed rest. Then, it was filtrated (Whatman No. 42 paper filter). A single drop (0.1%) of a sodium hexametaphosphate solution (NaPO_3_)_6_ was added to each extract. Conductivity was measured at 25 °C using a multi-parameter probe (Hanna Instruments Edge, Woonsocket, RI, USA).

Determination of heavy metals in soil samples was performed by inductively coupled plasma mass spectrometry analysis (ICP-MS, Agilent 7700 Series ICP-MS, Agilent Technologies, Tokyo, Japan). Firstly, dried soil was milled to a fine powder (granulometric fraction with diameter less than 2 mm) and then accurately weighed and mineralised. Mineralisation was performed by treating 500 mg of powdered samples with 9 mL of concentrated HCl and 3 mL of HNO_3_ and by heating (Ethos Touch Control, Milestone, Microwave Laboratory Systems) with a two-step procedure: 10 min to reach 200 °C followed by 15 min at 200 °C. Following cooling, the mineralised samples were transferred into vials, adjusted to a volume of 50 mL with MilliQ water, and diluted 25 times prior to ICP-MS analysis. This procedure was undertaken to ensure a maximum content of 5% acids and 0.2% dissolved solids. Quantification of all mineral elements was performed by ICP-MS. Standard reference materials (multi-element calibration standards 2A, Agilent) were used for precision, quality assurance, and checking the measurements. The average values of two replicates were taken for each analysis [58]. Total lead content was calculated as the mean of isotopic concentrations (^206^Pb, ^207^Pb, ^208^Pb). In order to evaluate pollutant plant uptake capabilities, heavy metal analyses on plant biomass (leaves and roots) samples have been performed according to the method reported in Angelini et al., 2022 [60].

### 4.5. Analysis of the Microbiological Soil Community

The total microbial abundance (N. cells/g soil) was determined by the epifluorescence direct count method with DAPI (4′,6-diamidino-2-phenylindole) as the DNA fluorescent dye. The DAPI staining method enables the detection of microbial cells in a sample, regardless their physiological state or metabolic activity. Soil samples (1 g each) were collected from each replicate microcosms and immediately transferred to a test tube containing 9 mL of a fixing solution (composed of phosphate-buffered saline: 130 mM NaCl; 7 mM Na_2_HPO_4_, 3 mM NaH_2_PO_4_; 2% formaldehyde (*v*/*v*); 0.5% Tween 20 (*v*/*v*) and 100 mM sodium pyrophosphate). After shaking, the suspensions were left to settle for 24 h to allow larger particles to precipitate. An aliquot of the supernatant was then treated with DAPI incubated 30 min in the dark at 4 °C and subsequently filtered through a 0.2 μm polycarbonate filter. Microbial cells were counted using a Leica DM 4000 B fluorescence microscope (Leica Microsystems GmbH, Wetzlar, Germany) [45].

The dehydrogenase activity (DHA), which reflects microbial respiration rate and provides information on the active portion of the soil microbial community [61,62], was evaluated in soil samples (6 g each) using a colourimetric method based on the quantification of 2,3,5-triphenyl formazan (TPF, red) produced from the reduction of 2,3,5-triphenylte-trazoliumchloride (TTC, uncoloured). Soil samples were incubated with TCC for 24 h at 37 °C in the dark. The soil microbial activity (expressed as TPF/g soil) was measured using a Multiskan Sky Microplate Spectrophotometer (Thermo Scientific, Waltham, MA, USA) [63]. All measurements were performed at the start (day 0) and at the end of the experiment (90 days).

### 4.6. Soil Quality Index (SQI)

To provide an integrated measure of soil functionality, a Soil Quality Index (SQI) was calculated integrating seven selected physicochemical and microbiological parameters: specifically, heavy metal concentrations (Zn, Sn, Pb), pH, electrical conductivity (EC), microbial abundance (DAPI counts), and dehydrogenase activity (DHA).

In accordance with [64], for each soil parameter we used, we calculated a parameter score (S) from 0 to 1 for each experimental condition, reflecting low and high quality, respectively. The two criteria we used are as follows:

More is better: S = (x − xmin)/(xmax − xmin).

Less is better: S = 1 − (x − xmin)/(xmax − xmin).

The more-is-better criterion was applied to pH, electrical conductivity, microbial abundance, and dehydrogenase activity for their positive influence for soil fertility. The less-is-better criterion was applied to heavy metal concentrations, because high concentrations can be toxic to soil organisms [65].

The overall parameter score (SQI) was calculated as reported by [66] as follows:$$SQI=\sum_{i=1}^{n}Si/n$$

In which *S* is each parameter score obtained from each parameter; *n*: number of parameters used (in our case, *n* = 7). The SQI can have values from 1 (the maximum quality) to 0 (the lowest quality).

### 4.7. Statistical Analysis

Statistical analyses were performed using R (v.4.3.0). The Shapiro–Wilk test was performed to verify the normality of the data distribution. Parametric (one-way or two-way ANOVA followed by Tukey HSD) or non-parametric (Kruskal–Wallis test followed by Wilcoxon post hoc tests) tests were applied on chemical and microbiological data.

For the multivariate analysis, a principal component analysis (PCA) was conducted on the standardised data (centred and scaled) to explore the relationships between chemical and microbiological variables and to visualise the influence of the treatments (presence of plant and/or microbial consortium) on the data distribution. The results of the PCA were represented graphically with confidence ellipses and group hulls. PERMANOVA was also performed to statistically test the effect of the experimental variables.

## 5. Conclusions

Overall, PLANT+BIOAUG was the most effective condition in promoting significant HM removal, and this was presumably due to both bioaugmented and natural soil microbial populations. This was possible thanks to the lavender rhizosphere, which favoured the co-existence of both bioaugmented and natural microbial populations, increasing both physical (through root development) and chemical (e.g., root exudate production) soil characteristics and its quality, as evidenced by the highest SQI value.

The combined application of *L. angustifolia* and a selected bacterial consortium represents a highly promising and functional approach to the sustainable management of contaminated soils, reducing heavy metals and stimulating overall soil microbial activity at the same time. These results were obtained with real contaminated matrices in microcosm conditions, and field applications are necessary in order to better evaluate the stability of the microbial consortium for a longer time and under variable conditions. Above all, competition with the native microbiota adapted to the variable site-specific conditions needs to be investigated. Knowing climate variability, water availability, root penetration, and pollutant amount and distribution in field conditions can be very useful for planning optimal inoculation strategies in actual environmental conditions and for stimulating bacterial consortium survival over time.

This approach is particularly effective because it aligns with the principles of agroecology and the circular economy. These principles combine environmental remediation with the utilisation of commercially valuable aromatic crops, providing a multifunctional solution for environmentally friendly contaminated site management.

## Figures and Tables

**Figure 1 plants-14-02734-f001:**
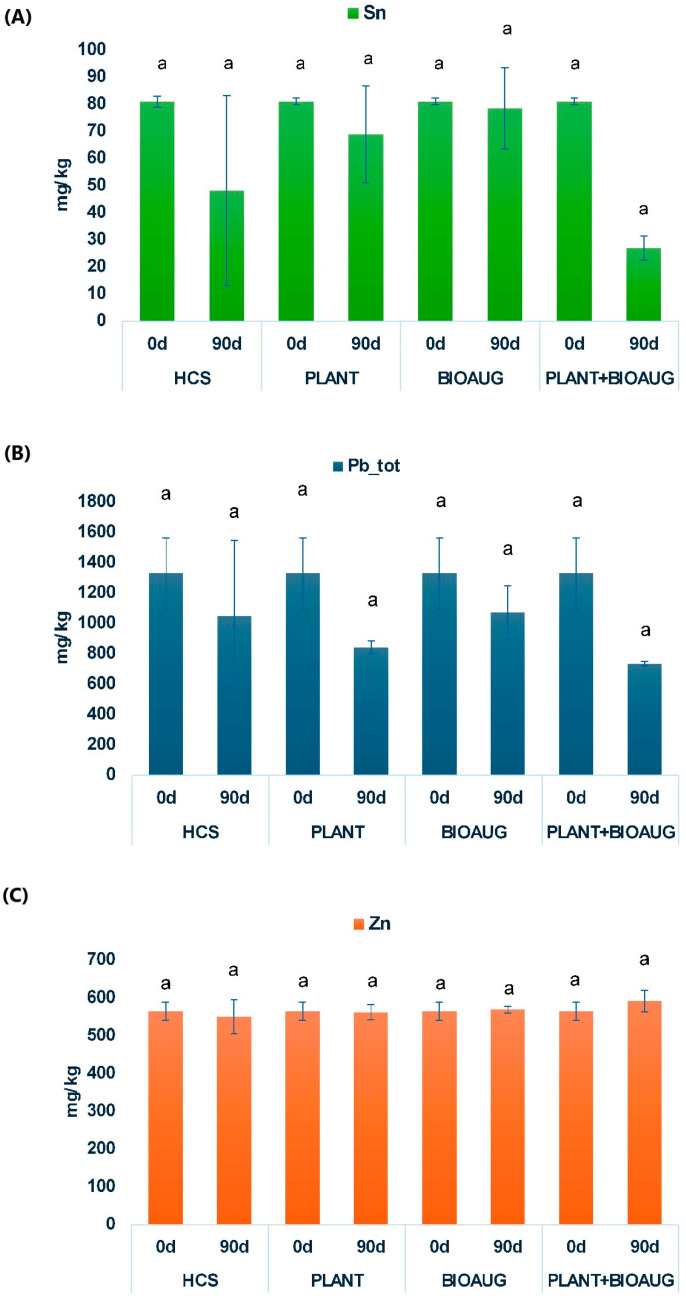
Average concentrations (mg/kg) of Sn (**A**), Pb (**B**), and Zn (**C**) in soils samples collected at 0 and 90 days in the various experimental microcosms (HCS, PLANT, BIOAUG, and PLANT+BIOAUG). Values are shown as mean ± standard error. Lowercase letter (a) is employed to evidence no statistical difference (Tukey’s post hoc test) among the experimental conditions.

**Figure 2 plants-14-02734-f002:**
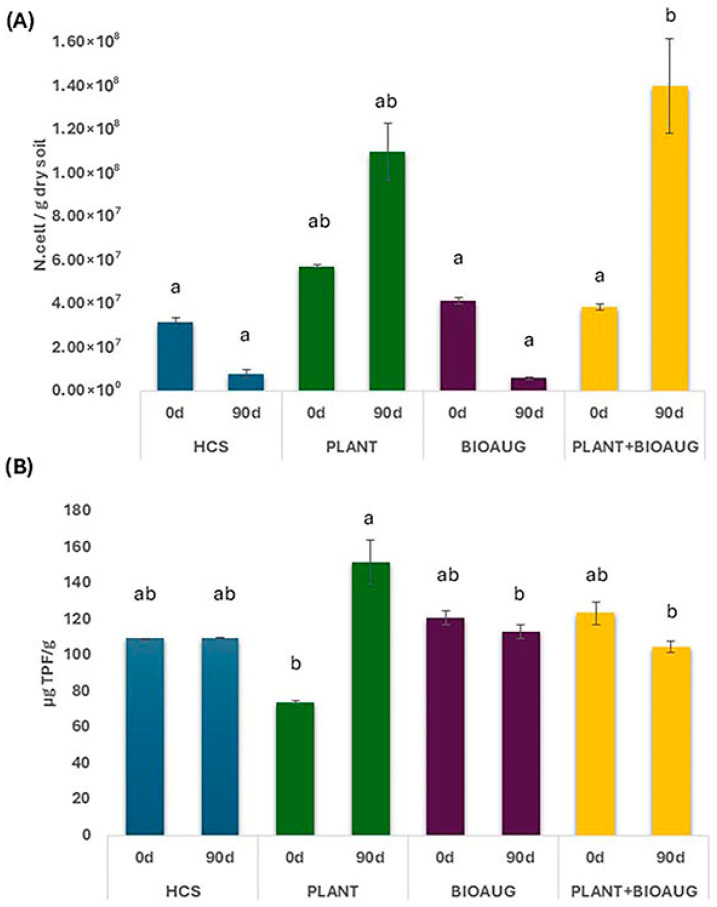
Average values of (**A**) total microbial abundance and (**B**) dehydrogenase activity in soils sampled after 0 and 90 days in the various experimental conditions (HCS, PLANT, BIOAUG, and PLANT+BIOAUG). Values are shown as mean ± standard deviation. Lowercase letters indicate statistically significant differences among the treatments.

**Figure 3 plants-14-02734-f003:**
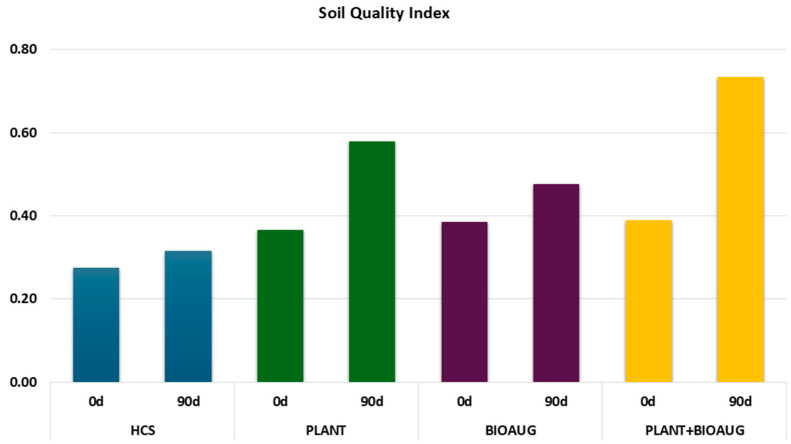
Soil Quality Index (SQI) values calculated by integrating pH, electrical conductivity, microbial abundance, dehydrogenase activity, and heavy metal parameters.

**Figure 4 plants-14-02734-f004:**
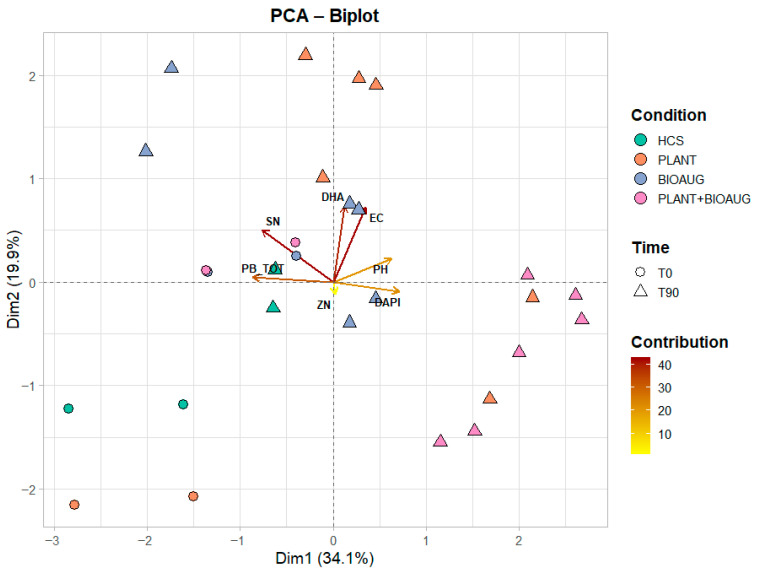
Principal component analysis (PCA) of physico-chemical and microbiological parameters (EC, pH, DHA, DAPI, and metal concentrations). The length and colour of each vector represent the distribution of the corresponding variable on the principal component (Dim1 and Dim2).

**Figure 5 plants-14-02734-f005:**
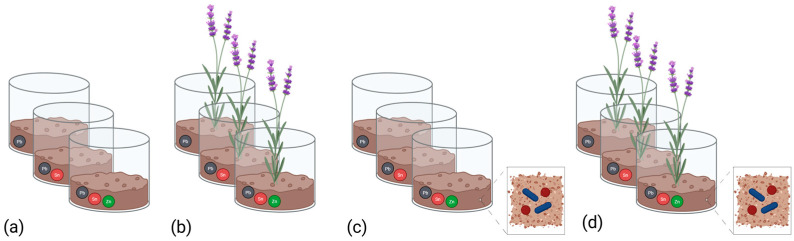
Experimental design of soil microcosms. (**a**): untreated contaminated soil (HCS), (**b**) soil planted with *Lavandula angustifolia* (PLANT), (**c**) soil inoculated with bacterial consortium (BIOAUG), and (**d**) combined treatment (PLANT+BIOAUG).

**Table 1 plants-14-02734-t001:** Soil pH and electrical conductivity (EC, dS/m) were measured at the commencement of the experiment and after 90 days under different treatment conditions. The values are expressed as the mean ± standard deviation among treatments and times. For pH results, superscript letters (*a*, *b*, *c*), indicating statistical differences in the application of Tukey’s post hoc test (*p* < 0.05), are reported.

Treatments	Days	pH	EC
		Mean ± st.dv	Mean ± st.dv
HCS	0	6.64 ± 0.23 ^*a*^	0.99 ± 0.01
	90	7.35 ± 0.08 ^*c*^	1.20 ± 0.04
PLANT	0	6.64 ± 0.23 ^*a*^	0.99 ± 0.01
	90	7.13 ± 0.00 ^*b*^	1.38 ± 0.18
BIOAUG	0	7.31 ± 0.06 ^*c*^	1.22 ± 0.01
	90	7.28 ± 0.04 ^*b,c*^	1.50 ± 0.08
PLANT+BIOAUG	0	7.31 ± 0.06 ^*c*^	1.22 ± 0.01
	90	7.38 ± 0.07 ^*b,c*^	1.39 ± 0.20

**Table 2 plants-14-02734-t002:** Zn, Sn, and Pb concentrations (average values and standard deviation) observed in plant tissues (leaves, roots) and TF (translocation factor) calculated for the different experimental conditions (PLANT, PLANT+BIOAUG).

		Zn	Sn	Pb
**PLANT**	*Leaves* *Roots* *TF*	37.15 ± 5.38 39.88 ± 16.67 0.93	0.10 ± 0.03 0.50 ± 0.02 0.20	5.74 ± 2.51 26.49 ± 6.59 0.22
**PLANT+BIOAUG**	*Leaves* *Roots* *TF*	11.26 ± 2.96 20.33 ± 11.53 0.55	0.10 ± 0.04 0.51 ± 0.03 0.20	15.46 ± 5.20 20.39 ± 6.21 0.76

## Data Availability

The data presented in this study are available on request from the corresponding author.

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
