# Peer review of "Synergistic Effects of *Lavandula angustifolia* and a Bacterial Consortium on Bioremediation of a Heavy Metal-Contaminated Soil"

_plants, 2025, doi:10.3390/plants14172734_

Round 1
Reviewer 1 Report
Comments and Suggestions for Authors
This work addresses a very interesting topic. It is not only interesting but also very important, as it concerns attempts to stabilize soils contaminated by industrial activity through the use of environmentally friendly methods.
The abstract is a good source of information about the research undertaken.
The keywords were well chosen.
The introduction provides a good overview of the problem addressed and considered in the manuscript and sufficiently supports the need for this research.
The methodology, description of results, and discussion are well described, although a few changes are needed.
The conclusions provide a good summary and indicate future research directions.
After analyzing the entire work, several comments and suggestions arise that should be considered before publication:
1. Description of results and discussion.
- In subsection 2.1., it would be worthwhile to add a table with the results obtained for individual objects and statistical data. After adding it, please renumber the tables already included in the current version of the work.
- In Figure 1, I propose presenting the Pb and Zn contents in separate figures, as is the Sn content.
- Please refer to the results obtained in Figure 1, which shows a significant decrease in Sn content in the control treatment after 90 days of incubation. Please describe this in the Results section and explain it in the Discussion section. The presented results indicate that Sn content decreases spontaneously. Additional treatments (PLANT or BIOAUG) are not necessary. The only significant effect is the PLANT+BIOAUG treatment. The situation is somewhat similar for Pb content, which also decreases after incubation without plants (PLANT) or BIOAUG treatments. Please explain these results.
- Table 1 requires supplementation with data from the BIOAUG treatment series.
- The description of the results presented in Table 1 is insufficiently comprehensive. Attention should be paid to the translocation coefficients.
- Line 209. Should be Figure 4. Please correct the notation below the figure.
2. Furthermore, please organize the literature cited in the text. Because the text cites references 1 to 37, then 46 to 57, and in the Materials and Methods section are references 38-45 cited, please move references 38-45 to the end of the reference list and then change the cited numbers in the text accordingly, according to the new order of the list.
3. In the references list, please remove duplicate reference numbers and revise the entire list according to the journal's requirements.
Author Response
Response to Reviewer 1 Comments
- Thank you very much for taking the time to review this manuscript “Microcosm experiments for evaluating synergistic effects of Lavandula angustifolia and a bacterial consortium for bioremediation of a heavy metal-contaminated soil”. Please find the detailed responses below and the corresponding revisions/corrections highlighted/in track changes in the re-submitted files.
- Quality of English Language
( ) The English could be improved to more clearly express the research.
(x) The English is fine and does not require any improvement.
|
Yes |
Can be improved |
Must be improved |
Not applicable |
|
|
Does the introduction provide sufficient background and include all relevant references? |
(x) |
( ) |
( ) |
( ) |
|
Is the research design appropriate? |
(x) |
( ) |
( ) |
( ) |
|
Are the methods adequately described? |
( ) |
(x) |
( ) |
( ) |
|
Are the results clearly presented? |
( ) |
(x) |
( ) |
( ) |
|
Are the conclusions supported by the results? |
(x) |
( ) |
( ) |
( ) |
|
Are all figures and tables clear and well-presented? |
( ) |
(x) |
( ) |
( ) |
- Point-by-point response to Comments and Suggestions for Authors
- Comments 1: In subsection 2.1., it would be worthwhile to add a table with the results obtained for individual objects and statistical data. After adding it, please renumber the tables already included in the current version of the work.
Response 1: In subsection 2.1, a table has now been added with the results obtained for individual objects and statistical data for pH (there are no significant differences for electrical conductivity - EC).
- Comments 2: In Figure 1, I propose presenting the Pb and Zn contents in separate figures, as is the Sn content.
Response 2: Figure 1 now presents the Pb and Zn contents in separate figures, as well as the Sn content.
- Comments 3: Please refer to the results obtained in Figure 1, which shows a significant decrease in Sn content in the control treatment after 90 days of incubation. Please describe this in the Results section and explain it in the Discussion section. The presented results indicate that Sn content decreases spontaneously. Additional treatments (PLANT or BIOAUG) are not necessary. The only significant effect is the PLANT+BIOAUG treatment. The situation is somewhat similar for Pb content, which also decreases after incubation without plants (PLANT) or BIOAUG treatments. Please explain these results.
Response 3: Thanks for your comment. We have improved Figure 1 and we added more explanations in the Results and Discussion sections (see lines 164-173 and 313-322 respectively).
- Comments 4: Table 1 requires supplementation with data from the BIOAUG treatment series.
Response 4: Table 1 shows the data for plant biomass, roots and leaves. Data for the BIOAUG treatment series are not shown, as that treatment did not include the plant component.
- Comments 5: The description of the results presented in Table 1 is insufficiently comprehensive. Attention should be paid to the translocation coefficients.
Response 5: Thanks for your kind suggestion. We have elaborated a new Table.
- Comments 6: Line 209. Should be Figure 4. Please correct the notation below the figure.
Response 6: We acknowledge the numbering issue and have corrected it.
- Comments 7: Furthermore, please organize the literature cited in the text. Because the text cites references 1 to 37, then 46 to 57, and in the Materials and Methods section are references 38-45 cited, please move references 38-45 to the end of the reference list and then change the cited numbers in the text accordingly, according to the new order of the list.
- Comments 8: In the references list, please remove duplicate reference numbers and revise the entire list according to the journal's requirements.
Response 7 and 8: The literature cited and references list in the text have been revised and reorganised.

Reviewer 2 Report
Comments and Suggestions for Authors
It is interesting and useful that the authors have investigated the effect of Lavandula angustifolia and a bacterial consortium on bioremediation of a heavy metal-contaminated soil. In general, the MS was written sound. Hence, it is recommended to be accepted after some revisions.
- Title
Change to “Synergistic effects of Lavandula angustifolia and a bacterial consortium on bioremediation of a heavy metal-contaminated soil”;
- Results
Add labels of significant difference in Figs. 1, 2 and 3.
- Discussions
Explain why the concentrations of Sn and Pb in soils significantly reduced after 90-days treatments, but not Zn?
- M&M
Delete Table 2;
- Conclusions
The paragraph (L414-424) should be deleted or moved to the section of discussion in somewhere if you think it is important.
Author Response
Response to Reviewer 2 Comments
- Thank you very much for taking the time to review this manuscript “Microcosm experiments for evaluating synergistic effects of Lavandula angustifolia and a bacterial consortium for bioremediation of a heavy metal-contaminated soil”. Please find the detailed responses below and the corresponding revisions/corrections highlighted/in track changes in the re-submitted files.
- Quality of English Language
( ) The English could be improved to more clearly express the research.
(x) The English is fine and does not require any improvement.
|
Yes |
Can be improved |
Must be improved |
Not applicable |
|
|
Does the introduction provide sufficient background and include all relevant references? |
(x) |
( ) |
( ) |
( ) |
|
Is the research design appropriate? |
(x) |
( ) |
( ) |
( ) |
|
Are the methods adequately described? |
(x) |
( ) |
( ) |
( ) |
|
Are the results clearly presented? |
( ) |
(x) |
( ) |
( ) |
|
Are the conclusions supported by the results? |
(x) |
( ) |
( ) |
( ) |
|
Are all figures and tables clear and well-presented? |
( ) |
(x) |
( ) |
( ) |
- Point-by-point response to Comments and Suggestions for Authors
- Comments 1: Change to “Synergistic effects of Lavandula angustifolia and a bacterial consortium on bioremediation of a heavy metal-contaminated soil”.
Response 1: The title of the manuscript has been improved following the suggestions.
- Comments 2: Add labels of significant difference in Figs. 1, 2 and 3
Response 2: Thank you for your suggestion, we have added labels of significant difference (a) in the figures (1-3).
- Comments 3: Explain why the concentrations of Sn and Pb in soils significantly reduced after 90-days treatments, but not Zn?
Response 3: Thanks for your question. We have add a sentence in the Discussion Section (see lines 324-327)
- Comments 4: Delete Table 2
Response 4: Table 2 in M&M has been delated.
- Comments 5: The paragraph (L414-424) should be deleted or moved to the section of discussion in somewhere if you think it is important.
Response 5: Thanks for your kind suggestion. All the authors agree to maintain this paragraph in the Conclusion section.

Reviewer 3 Report
Comments and Suggestions for Authors
-
Line 42–46: The introduction states that heavy metal pollution is a "threat to food security" but does not provide direct evidence or examples related to food crops in the study region. Consider adding specific citations or local case studies to strengthen this statement.
-
Line 72–74: The claim that combining lavender with biochar or bioaugmentation enhances metal stabilization would benefit from a brief mechanistic explanation or quantification from previous studies.
-
Line 93–103: The description of the bacterial consortium focuses on genera-level traits. It would be more informative to highlight any known synergistic effects between these genera when co-inoculated in similar environmental conditions.
-
Line 130–137: While pH changes are reported, the biological or chemical processes underlying these changes (e.g., root exudation, microbial metabolism) should be briefly discussed to support interpretation.
-
Line 143–148: In the contaminant removal results, statistical significance is indicated for Sn reduction, but the same analysis for Pb and Zn is not clearly explained. Please specify whether Pb reductions were statistically significant.
-
Line 154–158: The plant tissue metal concentration data are linked to prior work by Ancona et al., but the text does not critically compare similarities or differences with that study’s results.
-
Line 165–170: The microbial abundance increase in PLANT+BIOAUG is notable; however, possible causes for the sharp decrease in BIOAUG alone should be discussed earlier, not only in the discussion section.
-
Line 182–188: In the SQI section, the rationale for parameter weighting (equal weighting) is not justified. Could certain parameters be more critical for soil health than others?
-
Line 196–203: The PCA interpretation could be expanded by providing more biological meaning for the grouping patterns, rather than only noting statistical separation.
-
Line 217–224: The statement that plants and rhizosphere bacteria produce chelating compounds is plausible but should be supported with specific examples or references relevant to lavender.
-
The hypothesis that dead bacterial biomass stimulates native microbes is interesting. Consider supporting it with literature evidence or direct measurements of dissolved organic carbon.
-
The emphasis on root uptake reducing product contamination is important; however, the authors could link this to specific risk assessment frameworks or regulatory limits for aromatic plant products.
-
The bacterial consortium preparation method should include viability checks or cell counts post-centrifugation to ensure consistent inoculum quality.
-
For EC measurement, the use of sodium hexametaphosphate could affect ionic strength; please clarify whether this could influence EC readings and how it was controlled.
- Please read these two articles related to heavy metal contamination and include it in your introduction part (https://doi.org/10.1016/j.jhazmat.2024.136878), (https://doi.org/10.1016/j.scienta.2024.113575).
-
The conclusions recommend field application, but there is no preliminary scaling-up discussion. Suggest adding a paragraph on potential field constraints (e.g., water management, seasonal effects, soil heterogeneity).
The manuscript is generally well-structured and comprehensible; however, several sentences are long and contain multiple clauses, which occasionally reduces clarity (e.g., lines 83–92, 217–224). Simplifying sentence structures, and reviewing grammar for minor issues would improve readability and enhance the professional tone.
Author Response
Response to Reviewer 3 Comments
- Thank you very much for taking the time to review this manuscript “Microcosm experiments for evaluating synergistic effects of Lavandula angustifolia and a bacterial consortium for bioremediation of a heavy metal-contaminated soil”. Please find the detailed responses below and the corresponding revisions/corrections highlighted/in track changes in the re-submitted files.
- Quality of English Language
(x) The English could be improved to more clearly express the research.
( ) The English is fine and does not require any improvement.
|
Yes |
Can be improved |
Must be improved |
Not applicable |
|
|
Does the introduction provide sufficient background and include all relevant references? |
( ) |
(x) |
( ) |
( ) |
|
Is the research design appropriate? |
( ) |
(x) |
( ) |
( ) |
|
Are the methods adequately described? |
( ) |
(x) |
( ) |
( ) |
|
Are the results clearly presented? |
( ) |
(x) |
( ) |
( ) |
|
Are the conclusions supported by the results? |
( ) |
(x) |
( ) |
( ) |
|
Are all figures and tables clear and well-presented? |
( ) |
(x) |
( ) |
( ) |
- Point-by-point response to Comments and Suggestions for Authors
- Comments 1: Line 42–46: The introduction states that heavy metal pollution is a "threat to food security" but does not provide direct evidence or examples related to food crops in the study region. Consider adding specific citations or local case studies to strengthen this statement.
Response 1: References have been added to support the statements made in the introduction.
- Comments 2: Line 72–74: The claim that combining lavender with biochar or bioaugmentation enhances metal stabilization would benefit from a brief mechanistic explanation or quantification from previous studies.
Response 2: The references have been arranged in such a way as to clarify and demonstrate what has been stated.
- Comments 3: Line 93–103: The description of the bacterial consortium focuses on genera-level traits. It would be more informative to highlight any known synergistic effects between these genera when co-inoculated in similar environmental conditions.
Response 3: The authors would like to express their sincere gratitude to the reviewer for the thoughtful and insightful suggestions provided. The description of the bacterial consortium focuses on characteristics at the genus level, as there is no published evidence on the potential synergistic effects between these genera when inoculated together in similar environmental conditions. This is because the consortium was constructed in a previous study and tested only in the laboratory on organic contaminants (diesel).
- Comments 4: Line 130–137: While pH changes are reported, the biological or chemical processes underlying these changes (e.g., root exudation, microbial metabolism) should be briefly discussed to support interpretation.
Response 4: In the discussion section (lines 258-271), the biological or chemical processes underlying pH variations (e.g., root exudation, microbial metabolism) were briefly discussed to support the interpretation.
- Comments 5: In the contaminant removal results, statistical significance is indicated for Sn reduction, but the same analysis for Pb and Zn is not clearly explained. Please specify whether Pb reductions were statistically significant.
Response 5: The significance of Pb and Zn was also specified in the relevant section of the results (section 2.2. – lines 313-327).
- Comments 6: Line 154–158: The plant tissue metal concentration data are linked to prior work by Ancona et al., but the text does not critically compare similarities or differences with that study’s results.
Response 6: Thanks, we have added a sentence in the Discussion section (see lines 324-327)
- Comments 7: Line 165–170: The microbial abundance increase in PLANT+BIOAUG is notable; however, possible causes for the sharp decrease in BIOAUG alone should be discussed earlier, not only in the discussion section.
Response 7: In the results section (lines 197-200), the possible causes for the sharp decrease in BIOAUG treatment were briefly discussed.
- Comments 8: Line 182–188: In the SQI section, the rationale for parameter weighting (equal weighting) is not justified. Could certain parameters be more critical for soil health than others?
Response 8: The possibility to integrate overall parameters in the SQI made it possible to rank the soil improvement and to give a value of its quality in response to the remediation strategies applied. Really there are some parameters which are known drivers of soil quality such as contamination (in this case heavy metals), microbial abundance and dehydrogenase. Specifically, in our work, an increase in pH was a good result because make heavy metals less bioavailable in the soil. Regarding the EC because its variations were not significant did not have a substantial weigh for the index value.
We have added a sentence at lines 215-218 on the contribution of the parameters to the SQI.
- Comments 9: Line 196–203: The PCA interpretation could be expanded by providing more biological meaning for the grouping patterns, rather than only noting statistical separation.
Response 9: In the results section (lines 204-222), The PCA interpretation has been expanded by providing more biological meaning.
- Comments 10: Line 217–224: The statement that plants and rhizosphere bacteria produce chelating compounds is plausible but should be supported with specific examples or references relevant to lavender.
Response 10: We thank the reviewer for this valuable suggestion. In the revised manuscript, we have added specific references highlighting the ability of L. angustifolia to release chelating compounds, such as organic acids and phenolic metabolites, which contribute to the immobilization of heavy metals in soil. Accordingly, the paragraph has been modified to read as follows (lines 296-303).
- Comments 11: The hypothesis that dead bacterial biomass stimulates native microbes is interesting. Consider supporting it with literature evidence or direct measurements of dissolved organic carbon.
Response 11: We appreciate the reviewer’s insightful comment. In the revised manuscript, we have added references documenting that microbial necromass and cell lysis can release dissolved organic carbon and nutrients that stimulate the activity of native microbial communities (lines 281-286).
- Comments 12: The emphasis on root uptake reducing product contamination is important; however, the authors could link this to specific risk assessment frameworks or regulatory limits for aromatic plant products.
Response 12: We thank the reviewer for this observation. Products obtained from aromatic plants (such as essential oils) follow specific risk assessment frameworks and comply with European Regulations, which depend on their intended use. As we did not extract and evaluate essential oils from the plants under study, this assessment is not possible but it will be explored in further works.
- Comments 13: The bacterial consortium preparation method should include viability checks or cell counts post-centrifugation to ensure consistent inoculum quality.
Response 13: The method for preparing the bacterial consortium was carried out in the same manner as in the previously published work. The text (section M&M 4.2.) has undergone a process of refinement for the purpose of enhancing clarity.
- Comments 14: For EC measurement, the use of sodium hexametaphosphate could affect ionic strength; please clarify whether this could influence EC readings and how it was controlled.
- Response 14: Thanks for your comment. In this and in all our previous phytoremediation studies we have performed EC analyses according with the Italian official method for soil chemical analyses We have not registered any problems in EC readings. However, the overall amount of Na hexametophosphate (0.1%) solution employed for EC soil analyses was low, it consists in a single drop as described in the M&M section.
- Comments 15: Please read these two articles related to heavy metal contamination and include it in your introduction part (https://doi.org/10.1016/j.jhazmat.2024.136878), (https://doi.org/10.1016/j.scienta.2024.113575).
Response 15: We thank the reviewer for this useful suggestion. In the revised manuscript, we have incorporated both references in the Introduction section.
- Comments 16: The conclusions recommend field application, but there is no preliminary scaling-up discussion. Suggest adding a paragraph on potential field constraints (e.g., water management, seasonal effects, soil heterogeneity).
Response 16: Thanks for your comment. We think that adding a paragraph on potential field constrains, as you requested, needs a detailed and in-depth elucidation that can be done in a further work aimed to highlight and assess how water management, seasonal effects, soil heterogeneity and other important field constrains, can affect the efficiency of phytoremediation applications.
